

# Tritium activity concentration and behaviour in coastal regions of Fukushima in 2014

Michio Aoyama [1,2], Sabine Charmasson [3], Yasunori Hamajima [4], Celine Duffa [3]

Daisuke Tsumune[5], Yutaka Tateda[5]

[1] Centre for Research in Environmental Dynamics, University of Tsukuba, 1-1-1 Ten-noudai, Tsukuba 305-8572, Japan

[2] Institute of Environmental Radioactivity, Fukushima University, Kanayagawa 1, Fukushima 960-1296, Japan

[3] Institute de Radioprotection et de Sûreté nucléaire (IRSN), PSE-ENV, LRTA, BP3, 13115 Saint Paul lez Durance, France

[4] Institute of Nature and Environmental Technology, Kanazawa University, Wake, Nomi, Ishikawa 923-1224, Japan

[5] Environmental Science Research Laboratory, Central Research Institute of Electric Power Industry, 1646
Abiko, 270-1194, Japan

*Correspondence to*: Michio AOYAMA (michio.aoyama@ied.tsukuba.ac.jp)

**Abstract.** We observed [3]H activity concentrations and the [137]Cs activity concentrations during the SoSo 5 rivers cruise in 2014 and at the Tomioka port in 2014-2018. The [3]H activity concentrations at coastal stations located close to the Fukushima coast ranged from 90 Bq m$^{-3}$ to 175 Bq m$^{-3}$, and decreased between
67 Bq m$^{-3}$ to 83 Bq m$^{-3}$ at the stations located 12–16 km from the coast. The [3]H activity concentration at the estuaries and ports, except at 56 north canal of the FNPP1 site, are around 200-500 Bq m$^{-3}$ and slightly lower than the [3]H activity concentration of 500-600 Bq m$^{-3}$ observed in the rivers. These gradients of the [3]H activity concentrations in the coastal region might indicate the large effect of [3]H flux through the rivers. On the other hand, the [3]H activity concentration at 56N of the FNPP1 site was significantly high compared
to the [3]H activity concentration in surrounding waters both north and south of the FNPP1 site and in river waters. It should also be noted that the [3]H activity concentrations were similar at the stations located both north and south of the FNPP1 site, while the [137]Cs activity concentrations were lower at the stations north



of the FNPP1 site and higher at the stations south of the FNPP1 site. This indicated that the major sources of $^{137}$Cs could be the FNPP1 site as the point source while the source of $^3$H should be more diffuse and
linked to riverine inputs located north and south of the FNPP1 site. The $^3$H/$^{137}$Cs activity ratios in coastal waters were 1.2–2.2 as obtained via the slopes by standardised major axis regressions between the $^3$H activity concentration and the $^{137}$Cs activity concentration of SoSo samples and Tomioka Port observed in 2014-2018, which is significantly high compared to that of the released radionuclides derived from the FNPP1 site, which was 0.01 in 2011 just after the accident. The open-water $^3$H activity concentration
contribution to coastal waters was estimated to be $67 \pm 20$ Bq m$^{-3}$ and $66 \pm 17$ Bq m$^{-3}$ as the intercepts also by standardised major axis regressions. These estimates are consistent with 50 Bq m$^{-3}$ obtained at the Kuroshio region as the background levels of $^3$H activity concentration in open water. The $^3$H and $^{137}$Cs fluxes to the coastal region of Fukushima based on the open-water movement, freshwater flux from the rivers based on their respective catchment, and mean monthly precipitation were estimated. The largest $^3$H
flux is the open-water inflow from the north of the FNPP1 site and it reaches 52 GBq day$^{-1}$, while the rivers north of the FNPP1 site showed 3–5 GBq day$^{-1}$ fluxes. We obtained a $^3$H flux as 1.9–4.5 GBq day$^{-1}$ of $^3$H using the $^3$H activity concentration in the port, which is comparable with the fluxes obtained from the rivers located north of the FNPP1 site. While using $^3$H activity concentration at the 56 north canal of FNPP1, we obtained 28–86 GBq day$^{-1}$ fluxes, which is one order of magnitude larger than those estimated using  $^3$H
activity concentration in the FNPP1 port. One of the reasons could be the very high variability of the $^3$H levels at 56 north canal and in the port of FNPP1, explaining variable $^3$H/$^{137}$Cs activity ratio observed at 56 north canal and in the port of FNPP1. The $^3$H activity concentration of TFWT in the fish filets collected close to the FNPP1 site ranged from $97 \pm 11$ Bq m$^{-3}$ to $144 \pm 11$ Bq m$^{-3}$, which were similar to the $^3$H activity concentrations in the surrounding seawater, in agreement with the knowledge that the bioconcentration
factor of $^3$H is approximately 1. In contrast, higher values were found in TOBT, which can be linked to life-history traits.

## 1 Introduction

Tritium ($^3$H) is a radioactive isotope of hydrogen with a half-life of 12.4 yr. The major sources of $^3$H are cosmogenic, nuclear weapons tests, military production, and nuclear reactor operation, particularly heavy-





water reactor operation and nuclear fuel reprocessing plant (Galeriu and Melintescu, 2011).

Approximately 78 PBq ( PBq = $10^{15}$ Bq) of $^3$H is produced each year by cosmic ray spallation and the

inventory of $^3$H in the atmosphere is 1.3 EBq ( EBq = $10^{18}$ Bq); further, it was estimated that 240 EBq of

$^3$H was injected by nuclear weapons tests (Galeriu and Melintescu, 2011). From the nuclear fuel

reprocessing plants, 0.276 EBq was discharged mainly into the Atlantic ocean between 1970 and 2008

(Aoyama, 2019). In the nuclear reactor operations, during 1990–1997, 51.8 PBq of $^3$H was released into

the atmosphere, and 79.2 PBq of $^3$H was released into the ocean, as stated in Tables 32 and Table 35 in

annex C of the UNSCEAR 2000 report (UNSCEAR, 2000), most of which was released by heavy-water

reactors. Before the commencement of atmospheric testing of thermonuclear weapons in 1952, the $^3$H

content of precipitation was in the range of 180–1000 Bq m$^{-3}$ (Galeriu and Melintescu, 2013); this

background concentration resulted from cosmic ray spallation and showed a maximum at the mid-

latitudes of both the hemispheres. While elevated $^3$H levels have been measured in the atmosphere since

1952, $^3$H produced by thermonuclear testing has been the dominant source of $^3$H in precipitation

(Katsuragi et al., 1983;Galeriu and Melintescu, 2011).

The peak activity concentration of 201600 Bq m$^{-3}$ was recorded in the precipitation in Tokyo in the

northern hemisphere in March 1964 (Katsuragi et al., 1983). After the atmospheric nuclear weapons tests

were partly banned in 1963, the $^3$H levels in the precipitation began to decline gradually because of

radioactive decay and atmospheric dispersion, and was transferred into the ocean and groundwater via the

water cycle among the domains of atmosphere, ocean, and land with an apparent residence time of 23.0 ±

2.3 months (Katsuragi et al., 1983). In the southern hemisphere, $^3$H levels in the precipitation in Australia

were measured over the past 50 years (Tadros et al., 2014), demonstrating that the elevated levels of $^3$H in

the environment were due to last century's atmospheric thermonuclear testing. The peak in the $^3$H activity

concentration reached a maximum of 189000 Bq m$^{-3}$ (160 TU ) in 1963 owing to the peak in the

atmospheric tests, and then, from 1963 to present, a rapid drop in $^3$H activity concentration was observed.

This drop cannot entirely be from natural decay; therefore, it is attributed to the washout of $^3$H into the

oceans and groundwater. Since 1990, the levels of the $^3$H activity concentration have declined globally

and regionally; currently the levels of $^3$H in Australia are stable and in the range of 240–350 Bq m$^{-3}$ (2–3

TU), suggesting that today the $^3$H levels in the precipitation are predominantly due to naturally occurred





³H (Tadros et al., 2014). Some observations suggested that the effect of atmospheric weapon tests in the ³H activity concentration in the precipitation disappeared after 2000.

Further, explosions from the nuclear tests led to tritium emissions in the atmosphere as tritiated hydrogen (HT) and methyl tritium gas (CH$_3$T) were rapidly oxidised and converted into tritiated water molecules (HTO) that closely follow the whole water cycle and water masses (Ducros et al., 2018). Sugihara et al. (Sugihara et al., 2008) measured the ³H activity concentrations of 34 river waters in Japan in 2007–2008 and the ³H activity concentrations were 360–1660 Bq m$^{-3}$ (average 1060 ± 600 Bq m$^{-3}$).

They also reported that the ³H activity concentrations in the river waters showed a meridional distribution, and the ³H activity concentration was high at 35–43 °N and low in the south of 35 °N. In open-water, ³H was traditionally used as a tracer for ocean circulation, largely in the Pacific ocean (Michel and Suess, 1975;Van Scoy et al., 1991;Stark et al., 2004) and Sea of Japan (Kajì et al., 2005). Since the largest source of ³H in the environment atmospheric weapon tests, as stated previously, the ³H

activity concentration in the surface seawater showed a similar trend of freshwater on land, as expected. Therefore, a peak in the ³H activity concentration in the range of 500–12000 Bq m$^{-3}$ in the surface seawater was recorded in many places in 1963 in the northern hemisphere (Stark et al., 2004). The ³H activity concentration in the surface seawater decreased rapidly as along with the trend of the ³H activity concentration in the precipitation/river water. Povinec et al. (Povinec et al., 2013) estimated the pre-

Fukushima ³H activity concentration in the surface water in the north-western Pacific ocean as 70 ± 10 Bq m$^{-3}$ based on the data stored in the GLOMARD/MARIS database. In certain coastal regions, the ³H activity concentration is affected by the liquid and gaseous discharges from the nuclear fuel reprocessing plants (Aoyama, 2019;Masson et al., 2005) and the operation of nuclear power plants. Momoshima et al. (Momoshima et al., 1987) measured the ³H activity concentration in various environmental materials

around a typical nuclear power station, the Genkai Nuclear power plant, in Japan in 1983 and 1984. They found that the elevated ³H activity concentration was observed on one occasion in pine needles and surface soil after noting a high ³H activity concentration of 220–10000 Bq m$^{-3}$ in the seawater . However, no incidental increase in ³H levels was observed in atmospheric water vapour, hydrogen, and methane. Masson et al. (Masson et al., 2005) presented free-water ³H and organically bound ³H levels in the French

coastal marine environment, from Concarneau to Gravelines, along with the ³H levels in the seawater. The matrices selected for their specific survey included seawater, seaweed, molluscs, crustaceans, and





fish. The ³H activity concentration measured every month in the seawater at the Goury station, which is closest to the La Hague nuclear fuel reprocessing plant, in 2001–2002 varied from 3000 to 23000 Bq m⁻³, while in east Cotentin, the ³H activity concentration measured in the seawater along the French coast of

the English Channel was quite homogeneous and ranged from 3700 to 5900 Bq m⁻³.

They also reported the ³H activity concentrations of approximately 150 Bq m⁻³ south of Ireland and in the Atlantic Ocean. This activity concentration of 150 Bq m⁻³ is similar to, but slightly higher than, those reported in the north-western Pacific Ocean (70 ± 10 Bq m⁻³; Povinec et al., 2013) and may have been affected by the heavy discharges from the Europian fuel reprocessing plants of La Hague and Sellafield.

In this paper, we present results of the ³H activity concentration observed during the SoSo 5 rivers cruise and at the Tomioka port and Hasaki and discuss the behaviour of ³H in the coastal region of Fukushima. We also present results on the ³H contents found in the fish filet collected close to the FNPP1 site. These results are also discussed using the already published ³H activity concentrations of river and open-ocean waters.


## 2 Sampling and Method

### 2.1 Sampling

In October 2014, the SoSo 5 rivers cruise took place off the coast of the Fukushima Prefecture. The details of sampling radiocaesium have already been provided elsewhere (Aoyama et al., 2020 submitted).

During this cruise, 11 samples were collected for tritium measurement, 4 at the estuaries of the four rivers within 1 km from the river mouth, 4 at the stations within 1 km from the coast, and 3 at the stations 12-15 km from the coast (Aoyama et al., 2019; 2020a).

Since June 2014, the radiocaesium activity concentrations at the Tomioka port, which is located 10 km south of FNPP1, are being measured on a regular basis. However, limited ³H data were collected here. At

Hasaki, which is located 180 km south of FNPP1, the radiocaesium activity concentrations were measured on a regular basis, monthly to bi-monthly. The details of the sampling of radiocaesium are already provided elsewhere (Aoyama et al., 2012). Again, limited ³H data were collected here.



In October 2014, 11 samples of $^3$H were collected, 4 at the estuaries of the four rivers within 1 km from the river mouth, 4 at the stations within 1 km from the coast, and 3 at the stations 12-15 km from the coast

(Aoyama et al., 2019; 2020a).

In addition, two samples were collected at the Tomioka river in 2015 and 2018, and one sample was collected at the Tone river in 2015. All the sampling locations are shown in Figure 1 and Aoyama et al. (2021a, 2021b and 2021c).

### 2.2 Method

One litre of seawater collected in October 2014 during the SoSo 5 rivers cruise underwent $^3$H analysis through electrolytic enrichment at the $^3$H Laboratory (Miami, USA). The $^3$H activity concentration measurement of the seawater samples collected at the Tomioka port and Hasaki were also conducted through electrolytic enrichment at Chikyu Kagaku Kenkyusho Inc. (Nagoya, Japan), and the $^3$H activity concentrations of the fish filets were measured at the Kyushu Environmental Evaluation Association

(Fukuoka, Japan). We also measured the $^3$H activity concentration in the river waters of Tomioka and Tone rivers through electrolytic enrichment, too.

Fish filets collected at the Fukushima coast in 2014.were analysed with respect of Tissue-free water tritium, TFWT, which was extracted from the fish filet samples using freeze-drying techniques, where TFWT was trapped at low temperatures and then electrolytic condensation was performed to increase the

tritium activity concentration. Total organically bound tritium, TOBT, of the freeze-dried fish filet samples was obtained by complete oxidation of the organic compounds at high temperatures in an oxygen-rich environment, and then, the obtained tritiated water was trapped at low temperatures and then distilled. These distilled tritiated water samples were measured using a liquid scintillation counter.

### 3 Results


Tritium and dissolved $^{134}$Cs and $^{137}$Cs activity concentrations in the surface waters were obtained at several coastal stations of Fukushima and Ibaraki Prefectures during this study. These data are published as a dataset, entitled 'Dataset of $^3$H activity concentration, $^{134}$Cs and $^{137}$Cs activity concentrations in





dissolved form in surface water at several coastal stations of Fukushima and Ibaraki Prefectures during
the period from 2014 to 2018', doi: 10.34355/CRiED.U.TSUKUBA.00033 (Aoyama et al., 2020). The
tritium activity concentrations obtained during the SoSo 5 rivers cruises are already reported in a previous
study (Aoyama et al., 2019) and are also included in the dataset—doi:
10.34355/CRiED.U.TSUKUBA.00033.

The tritium activity concentrations in the fish filets collected at the Fukushima coast in 2014 are published
in a dataset, entitled 'Dataset of $^3$H activity concentration in fish filet collected at the Fukushima coast in
2014', doi: 10.34355/CRiED.U.TSUKUBA.00034 (Aoyama et al., 2020b).

The tritium activity concentrations in the river waters of Tomioka, Tone, and Ukedo rivers and the estuaries
of the four rivers between 2014 and 2019 are published in a dataset, entitled 'Dataset of $^3$H activity
concentration in river waters at Tomioka, Tone and Ukedo Rivers and estuary of four rivers during the
period from 2014 to 2019', doi: 10.34355/CRiED.U.TSUKUBA.00035 (Aoyama et al., 2020c).

## 3.1 $^3$H activity concentration in seawater

In 2014–2015, the $^3$H activity concentrations at coastal stations of Mano-1, Niida-1, Odaka-1, Uedo-1,
and Tomioka port ranged from 90 Bq m$^{-3}$ to 175 Bq m$^{-3}$, while the $^3$H activity concentrations at Niida-5,
Odaka-5, and Uedo-5 stations, which are located 11–15 km from the coastal stations, decreased and
ranged from 67 Bq m$^{-3}$ to 83 Bq m$^{-3}$ (Aoyama et al., 2020a). Further, the $^3$H activity concentration
obtained through NRC monitoring results within 20 km of the FNPP1 site also showed similar activity
concentrations as 110–190 Bq m$^{-3}$ in June 2014 and 65–1700 Bq m$^{-3}$ in October 2014. The $^3$H activity
concentration at Hasaki, which is located 176 km south of the FNPP1 site, ranged from 57 Bq m$^{-3}$ to 66
Bq m$^{-3}$ in late 2014 and early 2015. The TEPCO monitoring data at 56N of the FNPP1 site showed that $^3$H
activity concentration ranged from ND (ca. 1650 Bq m$^{-3}$) to 4300 Bq m$^{-3}$ between June 2014 and October
2014. Therefore, the $^3$H activity concentration at 56N of the FNPP1 site was significantly high compared
to that in the surrounding waters both north and south of the FNPP1 site, as shown in Figure 2. It should
be also noted that the $^3$H activity concentrations were similar at the stations located north and south of
FNPP1, while the $^{137}$Cs activity concentration was lower at the stations north of FNPP1 and higher at the
stations south of FNPP1. This indicated that the major source of $^{137}$Cs could be the FNPP1 site as a point



source, while the $^3$H might come from a broader range of locations since related to riverine inputs from rivers located both north and south of the FNPP1 site.

### 3.2 $^3$H activity concentration in fish filet

$^3$H activity concentrations of tissue free water, i.e. tissue-free water tritium (TFWT), in fish filet of *Sebastes cheni, Hexagrammos otakii, Okamejei kenojei, Lateolabrax japonicas, Paralichthys olivaceus,* and *Gadus microcephalus* vary from 97 ±11 Bq m$^{-3}$ to 144 ± 11 Bq m$^{-3}$ (Aoyama et al., 2020b), as shown in Figure 3. These fishes were collected close to the FNPP1 site and the $^3$H activity concentrations were similar to the $^3$H activity concentrations in the surrounding seawater, as shown in Fig. 3. This is consistent

with previous knowledge of rapid equilibrium between TFWT and the environment (Eyrolle et al., 2018;Eyrolle-Boyer et al., 2014). However, total organically bounded tritium (TOBT) exceeded TFWT with a ratio of TOBT to TFWT ranging from 1.54 to 2.10 in this study (Aoyama et al., 2020b), which indicates that these organisms fed on food that was contaminated earlier (Eyrolle-Boyer et al., 2014).

### 3.3 $^3$H activity concentrations in river water

The $^3$H activity concentration at 56 North canal, 56N, of Fukushima Dai-ichi Nuclear Power, hereafter FNPP1, varied significantly and showed a decreasing trend in general during the period from 2013 to 2019 while the $^3$H activity concentrations at Ukedo port and FNPP2 port were not significantly different during the period from 2013 to 2019 (Fig. 4).

As shown in Fig. 4, the $^3$H activity concentrations at Ukedo port and FNPP2 port were also lower in general rather than those at Ukedo river and Tomioka river before March 2015; however, recently, they became quite similar. This suggests a decreasing trend of the $^3$H activity concentrations at Ukedo and Tomioka rivers and a tendency of Ukedo and FNPP2 ports to retain almost constant $^3$H activity concentrations, as shown in Fig. 4. In 2014, we also observed that the $^3$H activity concentrations at

estuary and ports, except close to the FNPP1, are characterised by levels close to river water. Indeed, the $^3$H activity concentration ranged from 484 ± 17 Bq m$^{-3}$ in the Ota estuary to 214 ± 11 Bq m$^{-3}$ in the



Odaka estuary. The $^3$H activity concentration at Ukedo port was 330-840 Bq m$^{-3}$ and it was 300-480 Bq m$^{-3}$ at FNPP2 port. These levels are in good agreement with the $^3$H activity concentrations data reported by Ueda et al (2015)—ranging from 443 Bq m$^{-3}$ downstream of the Mano river to 600 Bq m$^{-3}$

downstream of the Ukedo river—and obtained by Fukushima Prefecture monitoring at Ukedo ranging between 590 and 650 Bq m$^{-3}$ in 2014. Further, the $^3$H activity concentration in coastal waters decreased rapidly with distance from the coast.

**4 Discussions**

**4.1 Relationship between $^3$H and $^{137}$Cs activity concentrations**

In 2014, the $^3$H activity concentrations in seawater at Tomioka port and SoSo 5 river samples showed a good linear relationship with the $^{137}$Cs activity concentration in seawater as shown in Figures 5 and 6 and Table 1, namely both radionuclides were added from the source(s) of which $^3$H/$^{137}$Cs activity ratio may be constant and may be difference at each region.

Takahata et al. (Takahata et al., 2018) reported high $^3$H and $^{137}$Cs activity concentrations (i.e. 80–290 and

80–13,800 Bq m$^{-3}$, respectively) in surface seawater sampled offshore from Fukushima shortly after the FNPP1 accident. The reported $^3$H concentrations were up to six times those of the pre-accident levels; however, the highest level was only approximately four times the background level attributed to global fallout. These concentrations are consistent with those reported by Povinec et al. (Povinec et al., 2013). Takahata et al. (Takahata et al., 2018) also reported a $^3$H/$^{137}$Cs activity ratio of $0.012 \pm 0.007$ for the

samples studied by them, which is similar to the $^3$H/$^{137}$Cs activity ratio in water that is released from the damaged FNPP1 reactors (~0.01; (Nishihara et al., 2015)).

Direct discharge of $^3$H into the ocean from FNPP1 was estimated to be approximately 0.05 PBq by multiplying the total $^{137}$Cs emission and the $^3$H/$^{137}$Cs ratio reported by Aoyama et al. (Aoyama et al., 2016b). This estimate ( $0.05 \pm 0.03$ PBq) is slightly lower than that obtained by Povinec et al. (2013)

using the same method ( $0.3 \pm 0.2$ PBq) for the samples collected in June 2011 from the south of the FNPP1 site during the KOK cruises (Buesseler et al., 2012).



In this study, we standardised major axis regression that considered both x-axis and y-axis errors to obtain a reliable $^3$H/$^{137}$Cs activity ratio from the mixing between open-ocean water and source waters from river and the FNPP1 site. As shown in Table 1, the $^3$H activity concentration of open-ocean water for the SoSo

samples was estimated to be $67 \pm 20$ Bq m$^{-3}$, while that for the Tomioka samples was $66 \pm 17$ Bq m$^{-3}$. This indicated that the $^3$H activity concentration in open-water might be approximately 70 Bq m$^{-3}$ in 2014 at the Fukushima coast. The $^3$H/$^{137}$Cs activity ratio in source waters was $2.2 \pm 0.9$ for SoSo cruise samples, and that for Tomioka port samples was $1.2 \pm 0.2$. The arithmetic mean of the $^3$H/$^{137}$Cs activity ratio at 56N of FNPP1 was ~7 during the period from May 2014 to March 2015, and this ratio was

extremely high compared with those of the SoSo cruise samples and Tomioka port samples. During the period from 2013 to 2016 at 56N of FNPP1, the $^{137}$Cs activity concentration decreased around two orders of magnitude lower due to decontamination effort of TEPCO while the $^3$H activity concentration decreased gradually, then the $^3$H/$^{137}$Cs activity ratio tended to increase from 1 to 10 during the period from 2013 to 2016 (Figure are not shown). This fact may consistent with lower the $^3$H/$^{137}$Cs activity ratio

observed in stagnant water (Nishihara et al., 2012) and in coastal waters close to FNPP1 (Povinec et al., 2013, Takahata et al., 2019).

## 4.2 $^3$H activity concentrations in river water and precipitation

Matsumoto et al. (Matsumoto et al., 2013) observed $^3$H activity concentrations in the precipitation samples that were collected after the FNPP1 accident, which occurred in Japan in 2011. Values exceeding

the pre-accident background were detected at three out of seven localities (Tsukuba, Kashiwa, and Hongo) southwest of the FNPP1 site at distances varying between 170 and 220 km from the FNPP1 source. The highest $^3$H content in the precipitation samples reached $19400 \pm 940$ Bq m$^{-3}$ ($164.2 \pm 8.0$ TU), which was 30 times higher than the pre-accident $^3$H level, and the $^3$H activity concentrations decreased steadily and rapidly with time, thereby becoming indistinguishable from the pre-accident

values within five weeks. $^3$H activity concentration in monthly precipitation in 2013–2014 were measured to be between 290 Bq m$^{-3}$ and 540 Bq m$^{-3}$ at Tamura-shi, Fukushima Prefecture located 35 km west from the FNPP1 site (Tagomori et al., 2015). Moreover, the $^3$H activity concentration in monthly precipitation in Niigata city in 2013–2014 were between $210 \pm 40$ Bq m$^{-3}$ and $890 \pm 80$ Bq m$^{-3}$ (Wang et al., 2016).





These values of the $^3$H concentrations might reflect cosmogenic $^3$H only and no significant atmospheric
release from the FNPP1 site to the atmosphere was observed during this time. It is also important to
notice that $^3$H content in precipitation during the typhoon period was lower than that in precipitation
during the non-typhoon period in the same month (Yamada et al., 2015). This observation might suggest
that the source of precipitation during typhoon should be of oceanic origin characterised by $^3$H levels
around 70 Bq m$^{-3}$ as indicated previously.

The Fukushima Prefecture government also monitors the $^3$H activity concentration in many rivers in
Fukushima. In 2013–2014, the $^3$H activity concentrations at Ukedo river and Tomioka river ranged from
< 650 Bq m$^{-3}$ to 1100 Bq m$^{-3}$ (Aoyama et al., 2020c), which were slightly higher than that in
precipitations (Tagomori et al., 2015) therefore, underground water of which the $^3$H activity concentration
might have been affected by the release of $^3$H from the FNPP1 site at the time of accident might impact
surface river water  leading to the increase in the $^3$H activity concentration in river water observed in
2013–2014. The $^3$H activity concentrations in these two rivers decreased gradually to approximately 300–
500 Bq m$^{-3}$ in 2019, which indicates that the $^3$H activity concentration in underground water might also
have decreased to a pre-accident $^3$H level without any effect from the FNPP1 accident. Nakasone et al.
(Nakasone et al., 2019) reported $^3$H in the monthly precipitation samples collected in Hokkaido, Gifu, and
Okinawa Prefectures between June 2014 and December 2017. The arithmetic mean of the $^3$H
concentrations in the collected precipitation samples for the period from June 2014 and December 2017 at
Hokkaido, Gifu, and Okinawa were estimated to be $620 \pm 270$ Bq m$^{-3}$, $302 \pm 120$ Bq m$^{-3}$, and $130 \pm 50$
Bq m$^{-3}$, respectively. These results exhibit well-known meridional distribution of $^3$H with latitude and
seasonal variations that is characterised by the highest and the lowest concentrations appearing in spring
and summer, respectively. It can be concluded that the $^3$H activity concentration in river water and in the
precipitation should have reach an equilibrium in 2018 and 2019. However, $^3$H releases from the FNPP1
site through probably underground water in this region led to higher activities in river waters in 2014 and
decreased afterward.



### 4.3 [3]H activity concentrations in open-ocean water and coastal waters off Fukushima

The background [3]H level for the western North Pacific Ocean has been estimated from the
      GLOMARD/MARIS database to reach up to $50 \pm 12$ Bq m$^{-3}$ (decay corrected to June 2011) in an area
      under the influence of the Kuroshio current (Povinec et al., 2017). Moreover, these levels could be 24 Bq
      m$^{-3}$ in an area under the influence of the Oyashio current (Watanabe et al., 1991). The [3]H activity
      concentrations in surface water in subtropical gyre obtained by the WOCE/GOSHIP hydrographic

program mainly along P2 and P3 lines in 1986, 1993, 1994, 2004, and 2013 are in a database WOD2018
      (Boyer et al., 2018) and the [3]H activity concentration was approximately 200–500 Bq m$^{-3}$ in 1986 while it
      decreased to ~50–60 Bq m$^{-3}$ in 2013 as shown in Fig. 7.

      Therefore in the subtropical gyre, such as Kuroshio region, the [3]H activity concentration in 2014 might be
      approximately 50 Bq m$^{-3}$, which is consistent with the evaluation of Povinec et al. (Povinec et al., 2013),

and also with [3]H activity concentration of about 70 Bq m$^{-3}$ observed in open-water by this study as shown
      in Table 4.

      In contrast, the [3]H activity concentration at coastal stations of FNPP1 obtained through monitoring  by the
      Japanese government and Fukushima Prefecture ( Fukushima Prefecture 1979, and its update until 2018)
      was approximately 1000 Bq m$^{-3}$ in 1986, as shown in Fig. 8 due to relatively large controlled liquid

discharges of [3]H from FNPP1 and FNPP2 in previous period while  the [3]H activity concentration just
      before the FNPP1 accident, i.e. 2002–2010, decreased to approximately 450–600 Bq m$^{-3}$, which indicates
      smaller controlled [3]H discharge of approximately 2–3 TBq year$^{-1}$ or 5–8 GBq day$^{-1}$ from both FNPP1 and
      FNPP2. After the FNPP1 accident, [3]H release could not be controlled from FNPP1 and were suspended
      from FNPP2, and not well mixed in the coastal region as shown in Figure 8, [3]H activity concentrations at

the station closest to the FNPP1 site were one order of magnitude higher than those at the other stations
      while no difference was observed before the accident among the monitoring stations.





### 4.4 Calculation of $^3$H and $^{137}$Cs fluxes at the Fukushima coast from rivers, open-ocean waters, and release from FNPP1


To discuss the contributions of various sources to the $^3$H flux to the ocean, we evaluated the $^3$H and $^{137}$Cs fluxes at the coastal region of Fukushima based on the open-ocean water movement from north to south at a speed of 0.05 m s$^{-1}$, freshwater flux from the rivers derived from the area of catchment surfaces of each river and the mean of monthly precipitation amount at five stations for precipitation observation along the

Fukushima coast. The details of assumptions and results for estimations of open water flux and fresh water flux by rivers are shown in Tables A1 and A2 in Appendix A.

We obtained the estimates for June 2014 and October 2014, and also with very heavy rains encountered in October 2019. The parameters for flux estimation are listed in Table 2, and the results of flux calculation are listed in Table 3.

Regarding the $^3$H fluxes, the largest source is from the open-water inflow from the north of FNPP1, and it reaches 52 GBq day$^{-1}$ while the rivers north of FNPP1 show 3–6 GBq day$^{-1}$ fluxes. From the port of FNPP1, we use Kanda's method (Kanda 2012), which considers the $^3$H activity concentration at Monoageba in the port and λ = 0.44 day$^{-1}$ for the exchange rate between the port and open water. Considering the volume of water in the port and λ, 9.6 m$^3$ s$^{-1}$ of water flux is estimated at the mouth of the

port, that led to the $^3$H fluxes in the range of 1.9–4.5 GBq day$^{-1}$ in three cases in 2014 and 2019, which is comparable with the $^3$H fluxes from the rivers located north of FNPP1. In contrast, considering Tsumune's method to estimate the flux from the FNPP1 site to the open-water by using the activity concertation at 56N of FNPP1 (Tsumune et al., 2012), the $^3$H flux from the FNPP1 site was found to be 28 to 86 GBq day$^{-1}$ as shown in Table 3, which is one order of magnitude larger than those estimated of 1.9–4.5 GBq

day$^{-1}$ from the port of FNPP1. One of the reasons could be the very high variability in the $^3$H results at 56N of FNPP1, which indicated a variable $^3$H/$^{137}$Cs activity ratio at 56N and the port of FNPP1.



## 5 Conclusion

In 2014–2015, the $^3$H activity concentrations at coastal stations Mano 1, Niida1, Odaka 1, Uedo 1, and Tomioka ranged from 90 Bq m-3 to 175 Bq m-3, and decreased between 67 Bq m-3 to 83 Bq m-3 at the

stations located 12–16 km from the coast. The $^3$H activity concentration at 56N of the FNPP1 site was significantly high compared to the surrounding waters both north and south of the FNPP1 site. It should also be noted that the $^3$H activity concentrations were similar at the stations located both north and south of the FNPP1 site, while the 137Cs activity concentrations were lower at the stations north of the FNPP1 site and higher at the stations south of the FNPP1 site. This indicated that major sources of 137Cs could be the

FNPP1 site as the point source while the source of $^3$H should be more diffuse and linked to riverine inputs located north and south of the FNPP1 site. The $^3$H activity concentration of TFWT in the fish filets of Hexagrammos otakii, Sebastes cheni, Okamejei kenojei, Lateolabrax japonicas, Paralichthys olivaceus, and Gadus microcephalus collected close to the FNPP1 site ranged from 97 ±11 Bq m-3 to 144 ± 11 Bq m-3, which were similar to the $^3$H activity concentrations in the surrounding seawater, in agreement with the

knowledge that the bioconcentration factor of $^3$H is approximately 1. In contrast, higher values were found in TOBT, which can be linked to life-history traits. The $^3$H/137Cs activity ratios derived from the land side were 1.2–2.2, which is significantly high compared to that of the released radionuclides derived from the FNPP1 site, which was 0.01 just after the accident. The open-water $^3$H activity concentration for the SoSo samples was estimated to be 67 ± 20 Bq m-3, and that for the Tomioka port sample was 66 ± 17 Bq m-3,

which is consistent with 50 Bq m-3 obtained at the Kuroshio region due to background levels of $^3$H activity concentration in open water. In 2014, the $^3$H activity concentration at the estuaries and ports, except at 56N of the FNPP1 site, are in the same range than river water. Indeed, the $^3$H activity concentration ranged from 484 ± 17 Bq m-3 in the Ota estuary to 214 ± 11 Bq m-3 in the Odaka estuary while the $^3$H activity concentration at the Ukedo port was 330–840 Bq m-3 and 300–480 Bq m-3 at the FNPP2 port. These levels

are in good agreement with the data obtained by Fukushima Prefecture monitoring within the Ukedo river, i.e. 590 and 650 Bq m-3, in 2014. Tthe $^3$H activity concentration in the coastal waters decreased rapidly offshore. The $^3$H and 137Cs fluxes to the coastal region of Fukushima based on the open-water movement, freshwater flux from the rivers based on the catchments of each river and mean monthly precipitation were estimated. The estimates were calculated for June 2014 and October 2014, and during very heavy rains



encountered in October 2019. The largest $^3$H flux is the open-water inflow from the north of the FNPP1 site and it reaches 52 GBq day-1, while the rivers north of the FNPP1 site showed 3–5 GBq day-1 fluxes. Regarding fluxes from the FNPP1 port, we used Kanda's method (Kanda 2012), based on $^3$H activity concentration at Monoageba in the port and the exchange rate between the port and open water. We obtained a $^3$H flux as 1.9–4.5 GBq day-1 of $^3$H, which is comparable with the fluxes obtained from the rivers located

north of the FNPP1 site. Using Tsumune's methods (Tsumune et al., 2012) to estimate the flux from the FNPP1 site to open water, we obtained 28–86 GBq day-1 flux, which are one order of magnitude larger than those estimated in this study using Kanda's method. One of the reasons could be the very high variability in the $^3$H levels at 56N and in the port of FNPP1, explaining variable $^3$H/137Cs activity ratio observed at 56N and in the port of FNPP1.

**Data availability**

Tritium activity concentrations in surface water and $^{134}$Cs and $^{137}$Cs activity concentrations in dissolved form in surface water at several coastal stations of Fukushima and Ibaraki Prefectures used in this study are provided in a published dataset, entitled 'Dataset of $^3$H activity concentration, $^{134}$Cs and $^{137}$Cs activity concentrations in dissolved form in surface water at several coastal stations of Fukushima and Ibaraki

Prefectures during the period from 2014 to 2018', doi: 10.34355/CRiED.U.TSUKUBA.00033 (Aoyama et al., 2021a).

Tritium activity concentrations in the fish filets collected at the Fukushima coast in 2014 used in this study are provided in a published dataset, entitled 'Dataset of $^3$H activity concentration in fish filet collected at the Fukushima coast in 2014', doi: 10.34355/CRiED.U.TSUKUBA.00034 (Aoyama et al., 2021b).

Tritium activity concentrations in river waters at the Tomioka, Tone, and Ukedo rivers and estuaries of four rivers between 2014 and 2019 used in this study are provided in a published dataset, entitled 'Dataset of $^3$H activity concentration in river waters at Tomioka, Tone, and Ukedo Rivers and estuary of four rivers during the period from 2014 to 2019', as doi: 10.34355/CRiED.U.TSUKUBA.00035 (Aoyama et al., 2021c).

**Author contribution**: SC, CD, and MA designed the experiment of SoSo 5 river cruise in 2014 and carried them out. MA designed the long-term experiment at the Tomioka port and Hasaki and carried them out. YH measured radiocaesium at Ogoya Underground laboratory for most of the samples considered in this



study. DT and YD collected the fish samples and prepared the fish filet for tritium measurements. MA designed the rivers water sampling, and MA and DT collected them.


**Competing interests**

The authors declare that they have no conflict of interest

**Acknowledgements**

This work was part of the AMORAD project (French State financial support managed by the National Agency for Research allocated in the 'Investments for the Future' framework programme under reference ANR-11-RSNR-0002). The authors thank the KANSO team as well as Mireille Arnaud, Franck Giner, and Hervé Thébault (IRSN) for their help and expert assistance in the fieldwork and preparation during the
cruise. The authors also thank Takashi Ishimaru (Tokyo University of Marine Science and Technology) and Takuji Mizuno (Fukushima Prefectural Fisheries and Ocean Research Centre) for their help in collecting and preparing the fish filet samples. The authors thank Toshiya Tamari (KEEA) for the measurement of the tritium activity concentrations in the fish filet samples as part of a contract with IER, Fukushima University. This work was also a part of a project funded by IER, Fukushima University to
Michio AOYAMA.

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





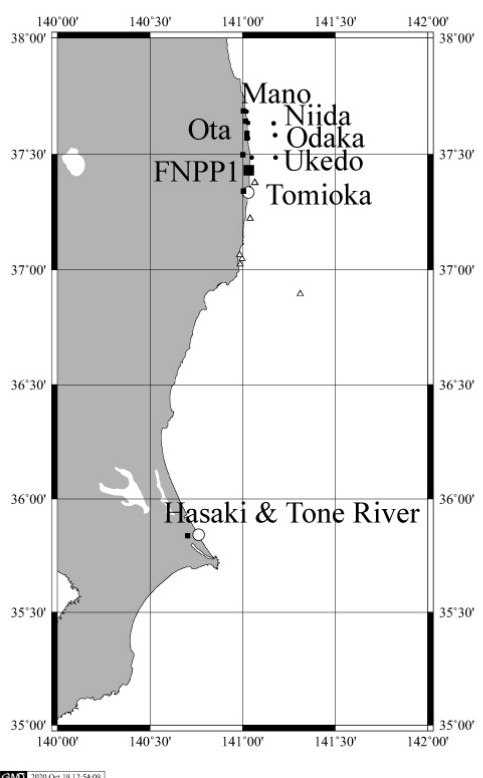

**Figure 1** Sampling locations.

Large solid squares: FNPP1 site and 56N of FNPP1

Large open circles: long-term observation at the Tomioka port and Hasaki

Small solid squares: Rivers and estuaries sampling site

Small solid circles: water sampling stations in October 2014

Open triangles: sampling locations of fish samples



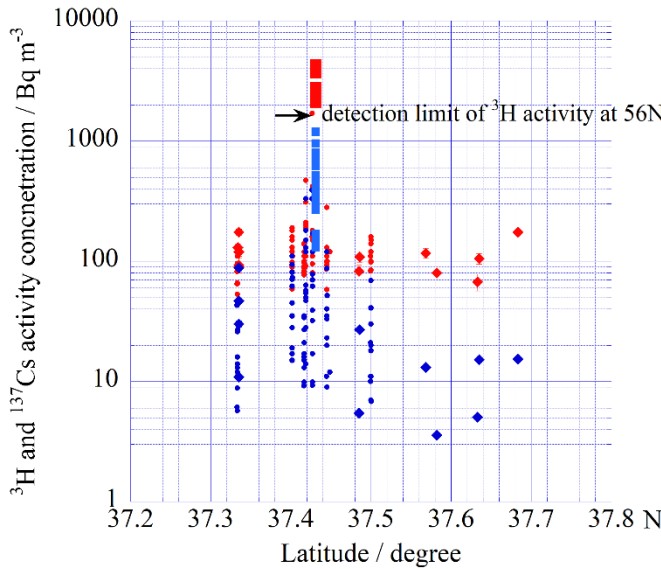

**Figure 2** Meridional distribution of the $^3$H and $^{137}$Cs activity concentrations along the Fukushima coast in 2014.

Red and blue squares: $^3$H and $^{137}$Cs activity concentrations, respectively, at 56N of the FNPP1 site. Several data of $^3$H activity concentration at 56N of the FNPP1 site were below the detection limit (ca. 1600 Bq m-3) and are not shown in this figure.

Red and blue diamonds: $^3$H and $^{137}$Cs activity concentrations, respectively, in the seawater samples collected during the 2014

SoSo 5 rivers cruise (this study).

Red and blue circles: $^3$H and $^{137}$Cs activity concentrations, respectively, in the long-term NRC monitoring of seawater samples (see text).





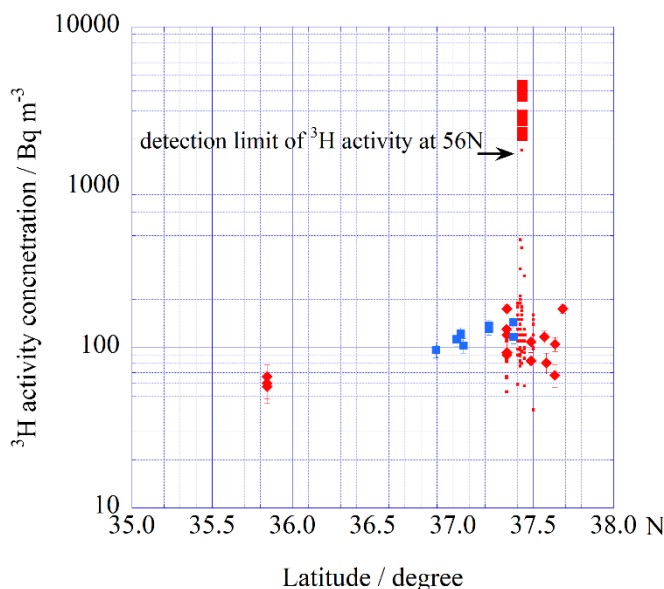

**Figure 3** Meridional distribution of $^3$H activity concentration in seawater and TFWT in fish filet along the Fukushima coast
in 2014.

Red solid triangle: $^3$H activity concentration in seawater samples (this study).

Blue solid square: TFWT in fish filet (this study).

Red solid square: $^3$H activity concentration in seawater samples at 56N of FNPP1 (large) and close to the FNPP1 site (small)

obtained by TEPCO and NRC monitoring, respectively. Several data of the $^3$H activity concentration at 56N of FNPP1 were

below the detection limit (ca. 1600 Bq m$^{-3}$) and did not appear in this figure.



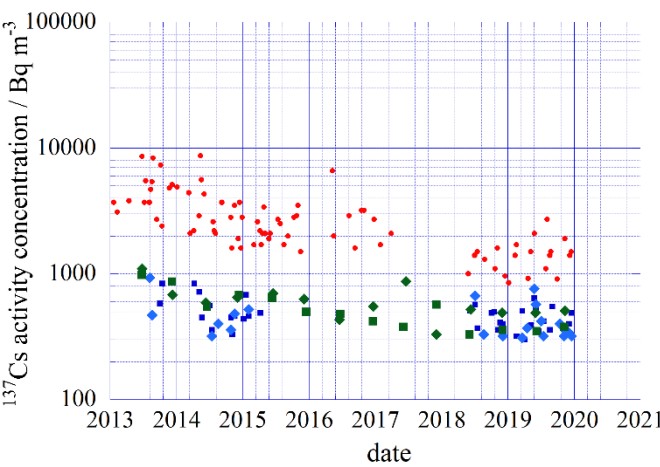


**Figure 4** Temporal changes in $^3$H activity concentrations in seawater and river water.

Red circles: $^3$H activity concentration at 56N of FNPP1 obtained through TEPCO monitoring (before March 2017, detection limit was ca. 1600 Ba m$^{-3}$).

Blue squares: $^3$H activity concentration at Ukedo port obtained through TEPCO monitoring.

Blue diamonds: $^3$H activity concentration at FNPP2 port obtained through TEPCO monitoring.

Green squares: $^3$H activity concentration at Ukedo River obtained through Fukushima Prefecture monitoring.

Green diamonds: $^3$H activity concentration at Tomioka river obtained through Fukushima Prefecture monitoring




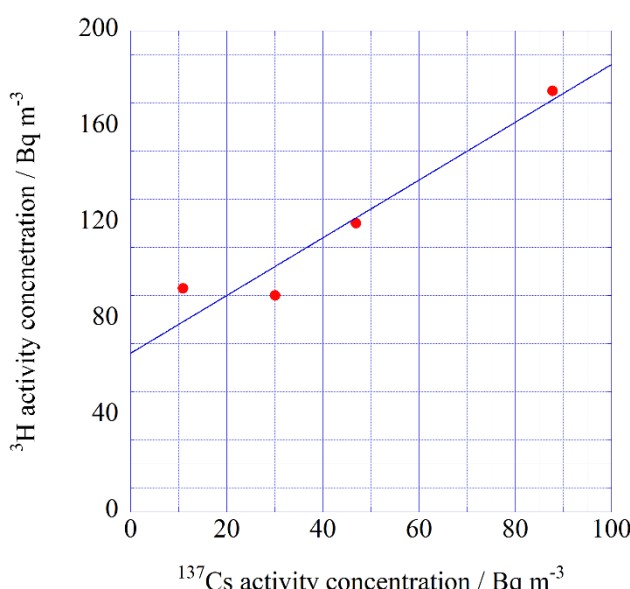

**Figure 5** Relationship between $^3$H and $^{137}$Cs activity concentrations in the samples collected from Tomioka port.

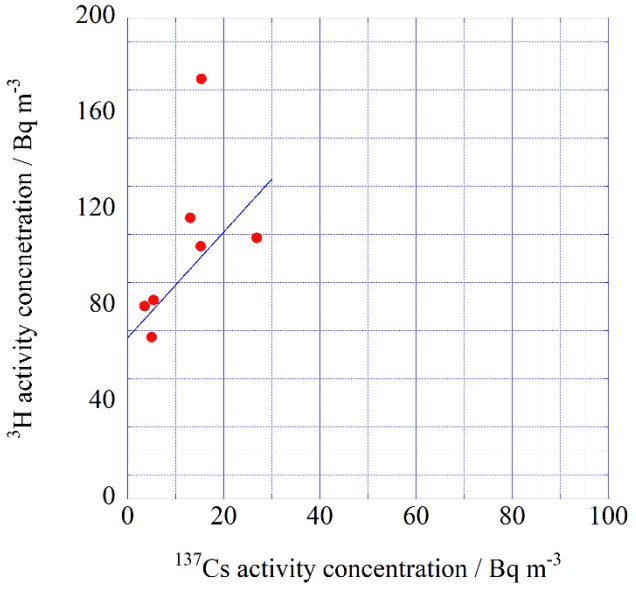

**Figure 6** Relationship between $^3$H and $^{137}$Cs activity concentrations in the samples collected from SoSo 5 river cruise.





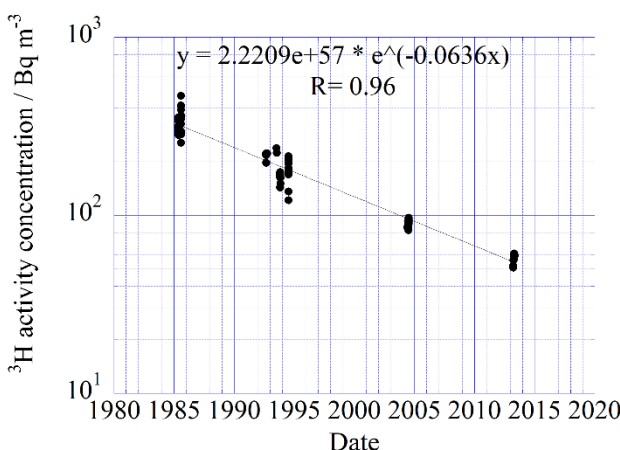

**Figure 7** ³H activity concentration in surface water in subtropical gyre in the North Pacific Ocean obtained through the

WOCE/GOSHIP hydrographic program.

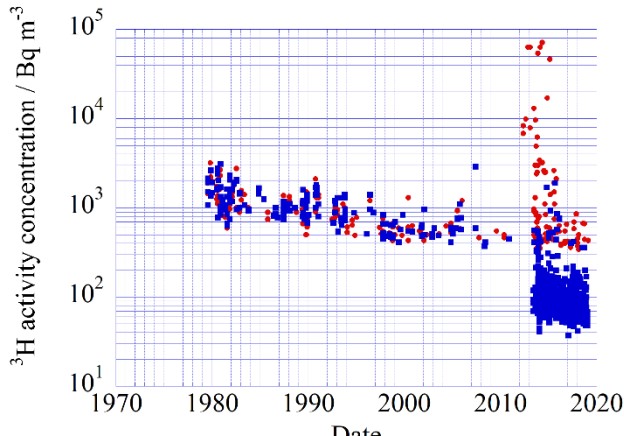

**Figure 8** ³H activity concentrations in surface water at the coastal region of Fukushima Prefecture from 1979 to 2019.

Red solid circle: ³H activity concentration in seawater samples obtained through NRC monitoring in front of the FNPP1 site.

Blue solid circle: ³H activity concentration in seawater samples obtained through NRC monitoring at the coastal region of

Fukushima Prefecture.





Table 1 Results of standardised major axis regression of $^{137}$Cs vs. $^{3}$H activity concentration

| Station/cruise | Intersept | | 3H/137Cs ratio | | | |
|---|---|---|---|---|---|---|
| | Bq m-3 | | | | | |
| SoSo cruise | 67 ± | 20 | 2.2 ± | 0.9 | October 2014 | This study |
| Tomioka port | 66 ± | 17 | 1.2 ± | 0.2 | June 2014 | This study |
| | | | | | | |
| 56N of FNPP1 | | | 7.4 | | May 2014 to March 2015 | TEPCO monitoring |
| | | | | | | |
| stagnant water | | | 0.01 | | March 2011 | Nishihara et al., 2012 |
| KOK cruise | | | 0.1-0.3 | | June 2011 | Povinec et al., 2013 |
| Close to FNPP1 | | | 0.012 ± | 0.007 | April 2011 | Takahata et al., 2018 |










Table 2 Parameters for flux estimation

|  | June 2014 | October 2014 | October 2019 |
|---|---|---|---|
|  |  |  |  |
|  | monthly precipitation (mm month-1) | | |
| The mean of 5 observation stations | 239 | 317 | 642 |
|  |  |  |  |
|  | 3H activity (Bq m-3) | | |
| open water | 50 | 50 | 50 |
| river Ukedo | 550 | 680 | 380 |
| 56N at FNPP1 | 4300 | 2800 | 1400 |
| In the port of FNPP1 | 2300 | 5420 | 1700 |
| river Tomioka | 590 | 650 | 510 |
|  |  |  |  |
|  | 137Cs activity (Bq m-3) | | |
| open water | 1.5 | 1.5 | 1.5 |
| river Ukedo | 230 | 230 | 60 |
| 56N at FNPP1 | 660 | 400 | 170 |
| In the port of FNPP1 | 2350 | 2500 | 1170 |
| river Tomioka | 230 | 230 | 60 |






Table 3 Estimated fluxes of $^{3}$H and $^{137}$Cs

| | June 2014 | October 2014 | October 2019 |
| --- | --- | --- | --- |
| | Bq day-1 | Bq day-1 | Bq day-1 |
| **Total 3H flux** | **5.8E+10** | **6.3E+10** | **6.1E+10** |
| Open water | 5.2E+10 | 5.2E+10 | 5.2E+10 |
| Rivers north of FNPP1 | 3.2E+09 | 5.3E+09 | 6.0E+09 |
| The port of FNPP1 | 1.9E+09 | 4.5E+09 | 1.4E+09 |
| Two rivers south of FNPP1 | 3.6E+08 | 5.6E+08 | 9.0E+08 |
| | | | |
| 56N at FNPP1(not included to total) | 8.6E+10 | 5.6E+10 | 2.8E+10 |
| | | | |
| **Total 137Cs flux** | **5.0E+09** | **4.6E+09** | **2.8E+09** |
| Open water | 1.6E+09 | 1.6E+09 | 1.6E+09 |
| Rivers north of FNPP1 | 5.9E+08 | 7.4E+08 | 1.9E+08 |
| The port of FNPP1 | 2.8E+09 | 2.1E+09 | 9.7E+08 |
| Two rivers south of FNPP1 | 1.2E+08 | 2.1E+08 | 5.4E+07 |
| | | | |
| 56N at FNPP1(not included to total) | 8.4E+09 | 8.0E+09 | 3.4E+09 |






Appendix A

Table A1. Open seawater flux.

| disnatnce from the coast | depth | section | speed | seawater flux |
|---|---|---|---|---|
| m | m | m2 | m s-1 | m3 day-1 |
| 2000 | 10 | 10000 | 0.05 | 4.3E+07 |
| 5000 | 25 | 62500 | 0.04 | 1.8E+08 |
| 10000 | 30 | 150000 | 0.032 | 3.8E+08 |
| 15000 | 50 | 375000 | 0.0256 | 4.4E+08 |
| | | | | |
| total | | | | 1.0E+09 |

Table A2. Fresh water flux by rivers located north and south of FNPP1

| River name | catchment* | portion to ocean | Precipitation in June 2014 | Fresh water flux | Precipitation in Oct. 2014 | Fresh water flux | Precipitation in Oct. 2019 | Fresh water flux |
|---|---|---|---|---|---|---|---|---|
| | km2 | | mm month-1 | m3 day-1 | mm month-1 | m3 day-1 | mm month-1 | m3 day-1 |
| Uda | 180 | 0.6 | 239 | 8.6E+05 | 317 | 1.1E+06 | 642 | 2.3E+06 |
| Mano | 168 | 0.6 | 239 | 8.0E+05 | 317 | 1.1E+06 | 642 | 2.2E+06 |
| Niiida | 261 | 0.6 | 239 | 1.2E+06 | 317 | 1.7E+06 | 642 | 3.4E+06 |
| Ota | 79 | 0.6 | 239 | 3.8E+05 | 317 | 5.0E+05 | 642 | 1.0E+06 |
| odaka | 67 | 0.6 | 239 | 3.2E+05 | 317 | 4.2E+05 | 642 | 8.6E+05 |
| Ukedo | 420 | 0.6 | 239 | 2.0E+06 | 317 | 2.7E+06 | 642 | 5.4E+06 |
| Maeda | 48 | 0.6 | 239 | 2.3E+05 | 317 | 3.0E+05 | 642 | 6.2E+05 |
| Total_North | 1223 | | | 5.8E+06 | | 7.8E+06 | | 1.6E+07 |
| | | | | | | | | |
| Kuma | 74 | 0.6 | 239 | 3.5E+05 | 317 | 4.7E+05 | 642 | 9.5E+05 |
| Tomioka | 63 | 0.6 | 239 | 3.0E+05 | 317 | 4.0E+05 | 642 | 8.1E+05 |
| Total_South | 137 | | | 6.5E+05 | | 8.7E+05 | | 1.8E+06 |
| *:catchments are cited from Sakuma et al., 2019 | | | | | | | | |