# Peer review of "Tritium activity concentration and behaviour in coastal regions of Fukushima in 2014"

_Biogeosciences, 2021_

## Author Comment (AC1)

**Replies to Comment on bg-2021-10 by anonymous referee #1**

**Dear anonymous referee #1,**

**We submit our replies to your comments as below.**
**Thank you very much for your comments which greatly contribute to the improvement of our manuscript.**

**Best regards,**

**Michio**
**\*\*\*\*\*\*\*\*\*\*\*\*\*\*\*\*\*\*\*\*\*\*\*\*\*\*\*\*\*\*\*\*\*\*\*\*\*\*\*\*\*\*\*\***

Comment:

This study estimated the tritium flux from rivers, Japan, and Fukushima Nuclear Power Plant after its incident caused by the tsunami in 2014 and during typhoon Hagibis and following typhoons in 2019.

However, the manuscript is partially difficult to read because a lot of information is missing and there are lots of typo. The contents of the text should be reconsidered. There are several topics which should be sufficiently discussed such as tritium in marine biota, difference in tritium between 2013-2014 and 2019, and the validation of the flux calculation and so on. All of the figures are very difficult to understand, particularly Figure 1 indicating sampling stations. I recommend that this figure is required to be revised.

Reply: The authors 'acknowledge that referee#1's comments will help improve this article, therefore we reply to the comments and will revise the article accordingly.
The all observed data are published as three assets of this article and many information are presented in the assets, therefore the authors did not repeat the information in the main text of this article. Referee #1 also stated that there are lots of typos, but the authors guess that referee#1 read this article as written in American English and stated "lots of typos". But, this article is written in British English throughout the main text.
Anyway, the authors have tried to improve our manuscript based on Referee#1 and #2's comment together with the authors' internal discussion.
To each comment of RC1, the authors respond as below.

L17 "Abstract."
I could not find out the information of the location of stations and areas the authors presented here. Because the authors described the concentration ranges at the specific stations (i.e. 56 north and soso 5 rivers) without any explanation, readers cannot distinctly understand what the authors intend to say. Define any abbreviations used in the abstract. This abstract seems to be kind of results and discussion section. As suggested in submission section of this journal (https://www.biogeosciences.net/submission.html), the abstract should be short, clear, and concise even though there is no limitation. This abstract must be short.

Reply:
The exact locations of the data in the abstract are already shown in the assets of this article

published as 10.34355/CRiED.U.TSUKUBA.00033, 10.34355/CRiED.U.TSUKUBA.00034 and 10.34355/CRiED.U.TSUKUBA.00035 and also most of the locations are in the current figure 1, too. The authors already prepare the revised figure 1 as appeared in this reply.

The abstract itself will be revised as below as requested by referee#1 and we also present our reply to each comment separately.

**Abstract:**
Tritium activity concentrations are reported in the coastal area close to the damaged Fukushima Daiichi nuclear power plant (FNPP1) as well as in estuaries and ports over the period 2014-2018. At distance around 15km from the coast, tritium activity concentrations get close to the levels which characterize open waters and which correspond to background levels (50-70Bq m-3). Within this coastal area levels ranged from 90 to 175Bq m-3 and seem to be mainly under the influence of riverine inputs. Indeed, the concentration of 3H activities in rivers were of the order of 500-600 Bq m$^{-3}$ which are values conventionally encountered at these latitudes and which do not allow to conclude as to a strong impact of the FNPP1 accident although levels in the canal discharge of reactors 5 and 6 north of the site clearly show higher tritium activities (up to 4300 Bq m-3). The contributions from various sources to the $^3$H flux to the coastal ocean were tentatively assessed. The largest 3H flux was the open-water inflow from the north of the FNPP1 site with 52 GBq day$^{-1}$ due to the large amount of seawater involved, while 9 rivers located north and south of FNPP1, together accounted for fluxes of 0.3 to 5 GBq day$^{-1}$. The FNPP1 port contributed between 1.9-4.5 GBq day$^{-1}$ while these fluxes reached up to 28-86 GBq day$^{-1}$ considering data in the discharge channel of the reactors 5-6 North of the FNPP1 site.

L17-18 "SoSo 5 rivers cruise in 2014"

What rivers did the authors indicate. Readers never know where those rivers are located. Reply: "SoSo-5rivers-cruise" means an observation campaign conducted in 2014 at Mano, Niida Odaka, Ota, and Ukedo rivers, and in this study the observed results of tritium activity concentration at four rivers among SoSo-5rivers-cruise except Ota river are used and also shown in 10.34355/CRiED.U.TSUKUBA.00033 with activity concentration data.

L18 "Tomioka port"

Reply: As indicated in current Fig.1, Tomioka port is located south of FNPP1 as shown "Tomioka". Now more clear in the revised figure 1.

L19 "Fukushima coast"

What coast is indicated? The Fukushima coast is composed by meridionally oriented coastline that is over 100 km long. Reply: as shown in the asset 10.34355/CRiED.U.TSUKUBA.00033, 90 Bq m-3 to 175 Bq m-3 are the data at Tomioka port, Mano 1 surface, Niida 1 surface, Odaka 1 surface and Ukedo 1 surface. These stations are located within 1km of the coast and are shown in figure 1. The exact locations by latitude and longitude are also in the asset 10.34355/CRiED.U.TSUKUBA.00033.

L21 "FNPP1"

Please specify this abbreviation.
Reply: Fukushima Daiichi Nuclear Power Plant = FNPP1

L21 "around 200-500 Bq m$^{-3}$"

This phrase seems to be redundant. " are in the range of 200-500"?
Reply: Yes, " are in the range of 200-500" is correct. " are in the range of 214+-11 Bq m-3 to 484+-17 Bq m-3" is a more correct statement based on data in the asset 10.34355/CRiED.U.TSUKUBA.00035.

L21-22 "slightly lower than"

Is that statistically appropriate?
Reply: as shown in the asset 10.34355/CRiED.U.TSUKUBA.00035, we show four data in the rivers in 2014 as 590, 650, 550 and 680 Bq m-3 while in the estuary we show also four data as 387+-13, 459+-15, 484+-17 and 214+-11 Bq m-3 in 2014. One-side t-test gave us that p= 0.09, therefore 3H activity concentration at the estuary is lower than those in river water at 91 % confidence level. So we stated slightly lower.

L24 "56N of the FNPP1"
This station is same as 56 north canal? Please specify.
I think that indicating concentration range is important because tritium flux from this area could affect tritium level in the surrounding "seawater", and is prevailing the riverine discharge.
Reply: "56N of the FNPP1" is the same place as 56 north canal, then we will use same words to 56 north canal in the revised main text.

L25 "both north and south of the FNPP1 site"
What areas did the authors indicate?
Reply: The area stated here is shown in current figure 3. The latitude range is from 37.3 N to 37.7 N with data at Hasaki at 35.84N.

L25-26 "in river waters."

What rivers did the authors indicate? SoSo 5rivers?
Reply: The 3H activity concentration data are shown in asset 10.34355/CRiED.U.TSUKUBA.00035 and the rivers are the Ukedo and Tomioka rivers.

L26-27 "at the stations located both north and south of the FNPP1 site"

Are those stations are different from above surrounding waters both north and south of the FNPP1? Please define the difference.
Reply: No, the authors stated the same data shown in figure 3.

L34-35 "The open-water $^{3}$H activity concentration contribution to coastal waters"

Does the phrase "open water activity concentration" refer to the activity concentration of tritium at cesium-137 = 0 on the x-axis? It seems difficult to understand from the abstract

alone, while the authors indicate intercepts by standardized major axis regressions.
Reply: No. In the open water, 137Cs originated from the atmospheric weapons tests and the level is around 1.5 Bq m-3. This sentence will be rewritten appropriately.

L37-39 "The $^3$H and $^{137}$Cs fluxes to the coastal region of Fukushima based on the open-water movement, freshwater flux from the rivers based on their respective catchment, and mean monthly precipitation were estimated."

From this concept, it is difficult to follow how to calculate the fluxes. In addition, there are no data on Cs-137 fluxes in this abstract.
Reply: Ok. This sentence will be rewritten appropriately.

L44-45 "which is one order of magnitude larger than those estimated using $^3$H activity concentration in the FNPP1 port."

Is this tritium flux at the FNPP1 port same as, "1.9-4.5 GBq day-1 $^3$H flux at $^3$H using the $^3$H activity concentration at the port"? Or is it Tomioka Port? Readers can be confused because one station has several different station names. The abstract also seems to have two different ports, so when the authors indicate a port, as " the port", the reader may not understand which port is referred to.
Reply: OK. This sentence will be rewritten appropriately.

L47-51 "The $^3$H activity concentration of TFWT in the fish filets collected close to the FNPP1 site ranged from 97 ±11 Bq m$^{-3}$ to 144 ± 11 Bq m$^{-3}$, which were similar to the $^3$H activity concentrations in the surrounding seawater, in agreement with the knowledge that the bioconcentration factor of $^3$H is approximately 1. In contrast, higher values were found in TOBT, which can be linked to life- history traits."

Discussion on the activity in fish has not conducted. I recommend that the authors delete it.
Reply: OK, we delete this.

L52 "1 Introduction"

The explanation concerning tritium in the environment is too long and is going out of focus on the local area of this study.
Reply: It is important to present the global tritium situation in order to understand tritium on a local scale. But the authors will shorten the introduction as you pointed out.

L103-107 "Momoshima et al. (Momoshima et al., 1987) measured the $^3$H activity concentration in various environmental materials around a typical nuclear power station, the Genkai Nuclear power plant, in Japan in 1983 and 1984. They found that the elevated $^3$H activity concentration was observed on one occasion in pine needles and surface soil after noting a high $^3$H activity concentration of 220–10000 Bq m$^{-3}$ in the seawater."

Again, this study focuses on the fluxes from rivers and port of the FNPP1, but the relationship between tritium level in seawater and that in marine organisms is not sufficiently discussed.
Reply: These statements links to Figure 8 in which the authors showed 3H activity

concentration close to nuclear power plant since 1979. We will delete these sentences because we will deleted Figure 8 and related sentences in the revised text.

L107-111 "However, no incidental increase in $^3$H levels was observed in atmospheric water vapour, hydrogen, and methane. Masson et al. (Masson et al., 2005) presented free-water $^3$H and organically bound $^3$H levels in the French coastal marine environment, from Concarneau to Gravelines, along with the $^3$H levels in the seawater. The matrices selected for their specific survey included seawater, seaweed, molluscs, crustaceans, and"

The same comment as above mentioned.
Reply: Ok, we will delete several of these sentences.

L120-121 "In this paper, we present results of the $^3$H activity concentration observed during the SoSo 5 rivers cruise and at the Tomioka port and Hasaki and discuss"

From the explanation given above by the authors, I do not think that the readers can relate the objectives of this study.
Reply: OK. The authors will explain more about the objectives of this study as shown below.
In this paper, we present results of the $^3$H activity concentration observed during the SoSo 5 rivers cruise and at the Tomioka port and Hasaki, a pier of the Hasaki Oceanographical Research Station of the Port and Airport Research Institute, and discuss the behaviour of $^3$H in the coastal region of Fukushima. An assessment of various contributions to both $^3$H and 137Cs fluxes into the coastal area close to the FNPP1 site was carried out taking into account i) riverine flux of these radionuclides based on precipitation amount on the catchments of several rivers, ii) fluxes estimation using from the FNPP1 site by previously reported method (Kanda 2012, Tsumune et al., 2012, 2013, 2020) and iii) fluxes from open water towards based on the speed of the coastal current and the activity concentrations characterizing these open water. All of these results are discussed together with 3H activity concentrations in river and open sea waters already published.

L120-121 "SoSo 5 rivers cruise and at the Tomioka port and Hasaki and discuss the behaviour of $^3$H in the coastal region of Fukushima."

Please specify. From figure 1, it is hard to find out where stations are. The authors should modify the station map of figure 1.
Reply: OK. The authors prepare a revised figure 1 as shown below. This figure will be further improved when the authors submit the revised manuscript.

[Figure]

L122-124 "We also present results on the $^3$H contents found in the fish filet collected close to the FNPP1 site. These results are also discussed using the already published $^3$H activity concentrations of river and open-ocean waters."

As commented in the abstract, there is no discussion of tritium levels in marine biota. I think that the authors have only discussed tritium flux from rivers and FNPP1, and made comparison of the flux from between river and the area in the proximity to the FNPP1. The authors should revise the objectives.
Reply: The authors will add discussions on 3H activity in fish and keep this issue as one of the objectives of the paper.

L177-179 "In 2014–2015, the $^3$H activity concentrations at coastal stations of Mano-1, Niida-1, Odaka-1, Uedo-1, and Tomioka port ranged from 90 Bq m$^{-3}$ to 175 Bq m$^{-3}$, while the $^3$H activity concentrations at Niida-5, Odaka-5, and Uedo-5 stations,"

It is difficult to understand where stations are located from Fig.1
Reply: OK. The authors prepared a revised figure 1 as shown above.

L179 "11–15 "

This range was 12-16 km in the abstract. Which is correct?
Reply: the range stated in the abstract is from the coastline and 11-15 km and in line 179 is from the first coastal stations of which distance from the coastline is around 1 km. Therefore both are correct. The authors will re-write to avoid confusion of the readers.

L186-187 "Therefore, the $^3$H activity concentration at 56N of the FNPP1 site was

significantly high compared to that in the surrounding waters both north and south of the FNPP1 site, as shown in Figure 2."

The comparison between data of 56N site with DL of <1600 Bq/m3 and that in other areas with DL of < 100 Bq/m3? is not appropriate because the mean value and concentration range can be different among 56N and other stations if DL is different between them.

Reply: We did not compare the mean values of two measurement groups, 56N and other areas, because we cannot calculate the exact mean values. But we know the total number of measurements as well as the numbers of data which were below the DL of <1600 Bq m-3 at 56N. As shown in Figure 2, it is clear that about half of the 3H activity data at 56N are higher than the 3H data obtained in other areas. Actually, at 56N there are 22 data with only 6 above DL and there are 84 data in other areas which were all above DL. Then, I put zero to the data below DL at 56N and did one side $t$-test. The $t$-test result shows that 3H activity concentration at 56N is higher than that in the other areas at a 98 % confidence level. A mean at 56N was 805+-1410 Bq m-3 (n=22) and that at the other areas was 147+-184 Bq m-3(n=84). So, it is safe to say that "Therefore, the 3H activity concentration at 56N of the FNPP1 site was significantly higher compared to that in the surrounding waters both north and south of the FNPP1 site, as shown in Figure 2."

L194-203 "$^3$H activity…"

The explanation on the activity concentration of tritium in fishes is only here. The authors should delete all the sentences related to this topic since it is difficult to relate the level of tritium in fishes to the main body of this study from only this information and less discussion.
Reply: Ok, we delete all the sentences related to this topic.

L206-207 "Fukushima Dai-ichi Nuclear Power, hereafter FNPP1, "
This specification should be done in the Introduction section.
Reply: Yes, it is.

L208 "FNPP2"

FNPP2? Please specify. This is not plotted on Fig 1.
Reply: FNPP2 is Fukushima Dai-ni Nuclear Power Plant. The authors added the location of FNPP2 in the revised figure 1.

L210-211 "in general"

What is "general" for? The difference can't be observed from Figure 4 since the both two shapes of plots are almost same from this resolution of Fig. 4. And sampling dates are different between river and port stations. Change of water/weather conditions due to different sampling date should have affected the tritium and Cs concentration in two sampling stations. Furthermore, the authors should check the difference in activity concentrations using statistical analysis.
Reply:
We performed a t-test on one side for three combinations as shown below which confirmed that our statements in the article are correct.
- In 2013-2014, the average of the 3H activity concentrations observed in the Ukedo and Tomioka rivers (762 Bq m-3; n = 8) was higher than that observed at the ports of Ukedo and FNPP2 (534 Bq m -3; n = 16) at a confidence level of 99%.

- At the Ukedo and FNPP2 ports, the average of the 3H activity concentrations was higher in 2013 and 2014 (534 Bq m-3; n = 16) compared to 2018-2019 (average of 426 Bq m-3; n = 37) at a confidence level of 99%.
- In 2018-2019, the averages of the 3H activity concentrations in the Ukedo and Tomioka rivers (429 Bq m-3; n = 8) and at the Ukedo and FNPP2 ports (426 Bq m-3; n = 37) do not show a significant difference according to the t-test on one side with p = 0.48

L239-241 "This estimate ( 0.05 ± 0.03 PBq) is slightly lower than that obtained by Povinec et al. (2013) using the same method ( 0.3 ± 0.2 PBq) for the samples collected in June 2011 from the south of the FNPP1 site during the KOK cruises (Buesseler et al., 2012)."

I do not think that 0.05 PBq is slightly lower than 0.3 PBq. The explanation of the difference is necessary.
Reply: Although one of the authors, M. Aoyama, is one of the authors of the article by Povinec et al., 2013, but not the author of the article by Takahata et al., 2018. Therefore we do not have a responsibility to explain the difference in the results from the two articles. The appropriate statement is "This estimate ( 0.05 ± 0.03 PBq) is  lower than that obtained by Povinec et al. (2013) using the same method ( 0.3 ± 0.2 PBq) applied on the samples collected in June 2011  south of the FNPP1 site during the KOK cruise (Buesseler et al., 2012)." And we can add  some lines as below.
 Povinec et al. used 3H/137Cs activity ratio and estimated 137Cs released amount from both direct discharge and atmospheric deposition for a wide area from the samples collected in June 2011 south of the FNPP1 site during the KOK cruise (Buesseler et al., 2012), while Takahata et al., used 3H/137Cs activity ratio observed very close to FNPP1 site and the direct discharge amount of 3.5 +-0.07 PBq estimated by Tsumune et al., 2012. So the estimated amunt of tritium release from FNPP1 site depend both 3H/137Cs activity ratio and estimated amount of 137Cs..

L250-254 "During…. 10 during the period from 2013 to 2016 (Figure are not shown)."

A figure showing the decline of the ratio with time is necessary. A detailed explanation of the reason is highly required because there could be process causing fraction between Cs and tritium during the water movement, other than the decontamination effort of TEPCO. Salinity change could also affect the ratio, however there were no description concerning the change in the ratio along the salinity gradient.
Reply: Ok, both 3H/137Cs activity ratio and their respective activities during the period 2012 -2016 are showed in figures below.  3H/137Cs activity ratio increase over time as mentioned in our text and not decline, while activity concentrations of 3H and 137Cs at 56N, shows a different decreasing trend.

[Figure]

[Figure]

In the current text, we stated "During the period from 2013 to 2016 at 56N of FNPP1, the 137Cs activity concentration decreased around two orders of magnitude lower due to decontamination effort of TEPCO while the 3H activity concentration decreased gradually, then the 3H/137Cs activity ratio tended to increase from 1 to 10 during the period from 2013 to 2016 (Figure are not shown). " Only we need to do here is to revise from "two orders" to "one order". We keep the other part of the sentence because it is the correct description of the trend of the two radionuclides. Since the observation at 56N was carried out very close to the source, we can assume that the observed trend was little affected by the water movement but mostly represents the change in the characteristics of the source. If the observation was done after some movement, salinity may affect the ratio as referre 1 pointed out, but this is not the case here.

So, in the revised main text, this part will be as below;
Between 2013 and 2016 at 56N of FNPP1, the 137Cs activity concentration decreased by about one order of magnitude due to decontamination effort of TEPCO while the 3H activity concentration decreased only gradually, leading to an increase of the 3H/137Cs activity ratio from 1 to 10 (Figure are not shown).

L254-256 "This fact may consistent with lower the $^{3}$H/$^{137}$Cs activity ratio observed in stagnant water (Nishihara et al., 2012) and in coastal waters close to FNPP1 (Povinec et al., 2013, Takahata et al., 2019)."

I think this sentence is grammatically wrong.
Reply: One key number about $^{3}$H/$^{137}$Cs activity ratio just after the accident was  0.012+-0.007

for the seawater collected around 30km from FNPP1 in May 2011 by Takahata et al., 2018 and around 0.01 in the stagnant water collected in March and April 2011 by Nishihara et al., 2012 and 2015. Povinec et al., 2013 also reported that $^3$H activity concentration ranged from 50 to 150 Bq m-3 at KOK cruise area (Buesseler et a., 2012) while 137Cs activity concentration was 10 – 3000 Bq m-3 indicating 3H/137Cs activity ratio was less 1 when subtracting background activity of 3H as around 50 Bq m-3.

Therefore, we revised this sentence to  clarify the meaning and correct the grammar as below.
 "This low 3H/137Cs activity ratio observed in 2012 by TEPCO is consistent with low 3H/137Cs activity ratio reported in stagnant water as around 0.01 (Nishihara et al., 2012 and 2015) and in coastal waters close to FNPP1 as 0.012+-0.007  in May 2011 ( Takahata et al., 2019)."

 L258 "Matsumoto et al. (Matsumoto et al., 2013)" Is
 that correct reference style? Please check.

Reply: Ok. We will correct if we keep this reference in the revised text though we may delete this.

 L260-261 "Tsukuba, Kashiwa, and Hongo"

 Indicating location and values on figure would be easy to understand for readers.

Reply: To simplify the discussion about 3H activity concentration in the precipitation, the authors decided to delete lines 258-274.

 L266 "Tamura-shi"

 Same as above comment

Reply: We put a mark at Tamura-shi in the revised figure 1.

 L268 "Niigata city"

 Same as above comment

Reply: To simplify the discussion about 3H activity concentration in the precipitation, the authors decided to delete line258-274 in section 4.2 as below.

**4.2 $^3$H activity concentrations changes in river water**

The Fukushima Prefecture government also monitors the $^3$H activity concentration in many rivers in Fukushima (https://www.pref.fukushima.lg.jp/site/portal/ps-kasensui-tritium-kako.html accessed on 20 October 2020 in Japanese only). In 2013–2014, the $^3$H activity concentrations at Ukedo and Tomioka rivers ranged from 550 Bq m$^{-3}$ to 1100 Bq m$^{-3}$, which were slightly higher than that in precipitations observed at Tamura-city (Tagomori et al., 2015, https://keea.or.jp/pdf/knakyokanri/44/vol_44_08.pdf, accessed on 9 October 2020).  Therefore, underground water, of which the $^3$H activity concentration might have been affected by the atmospheric release of $^3$H from the FNPP1 site at the time of accident, might impact the increase in the $^3$H activity concentration in river water observed in 2013–2014. The $^3$H activity concentrations in these two rivers decreased gradually to approximately <330-510 Bq m$^{-3}$ in 2019, which indicates that the $^3$H activity concentration in underground water might also have decreased to a pre-accident $^3$H level without any further effect from the FNPP1 accident.

 L269 "no significant"

Did the authors check statistically?

L273-274 "by $^3$H levels around 70 Bq m-3 as indicated previously."
Was the level consist with the observed values during the typhoon period?
Reply:3H activity concentration in precipitation during the typhoon period is very low and reached 90+-10 Bq m-3 (Yamada et al., 2015, Radioisotopes). But this part will be deleted in the revised article.

L277 "< 650"
550 is minimum value from Aoyama et al., 2021c. Why did the authors describe like <650? All readers wrongly recognize that DL is about 650.
Reply: Just a typo. "In 2013–2014, the 3H activity concentrations at the Ukedo and Tomioka rivers ranged from 550 Bq m−3 to 1100 Bq m−3 (Aoyama et al., 2021c)," is a correct statement.

L277 "(Aoyama et al., 2020c), "

This is not listed in Ref.
Reply:   This is Aoyama et al., 2021c and in the reference list.

L277 "which were slightly higher than"

The authors should carry out statistical analysis.
Reply: as we stated previously, the one-side t-test gave us that p= 0.09, therefore 3H activity concentrations in river water is higher than those at the estuary at 91 % confidence level. Therefore, we think that slightly higher should be appropriate.

L277-278 "that in precipitations"

Were those values obtained from the same area? The minimum concentration by Fukushima Pref was over 300 Bq/m3. Since the detection limit of the tritium analysis conducted by Fukushima Prefecture could be higher than that by Tagomori et al., it may be impossible to make an appropriate comparison.
Reply: No, we compare the 3H activity concentration in precipitations and river waters in 2014, and all values were above DL in 2014, allowing us to compare them.

L281-282 "approximately 300– 500"

<330-510 is correct range.
Reply: Yes, <330-510 is the correct statement for the range. The authors will re-write this sentence.

L287 "Hokkaido, Gifu, and Okinawa

Same as other areas: Please indicate location.

I do not know where those are located. For example, from the map, Hokkaido is the name of a province whose area exceeds 80,000 km2. Tritium levels are dependent on sampling points, so information on sampling location may be necessary.

Reply: I deleted the sentences where we cite these data.

L290-293 "that the 3H …. decreased afterward. "

The authors should conclude this trend from the statistical analysis. For example, was there significant difference in tritium between 2014 and 2018-2019?

Reply: The average of the 3H activity concentrations in the Ukedo and Tomioka rivers were significantly higher in 2013 and 2014 (762 Bq m-3; n = 8) compared to 2018 and 2019 (429 Bq m-3; n = 8) with a 99% confidence level.

L297 "24 Bq"

It is too low. 240 Bq/m3?

Reply: No, this value is correct.

L298 "Watanabe et al., 1991" This study is not cited in Ref

Reply: Ok. We added this in the reference.

L306 "in Table 4"

Table 4 is not listed in this manuscript. Or estimated values from the intercept in Table 1?

Reply: Just a typo. Table 1 is correct as you guess.

L311-313 "before the …or 5–8 GBq day-1 from both FNPP1 and FNPP2. "
Please explain how the authors estimated the amount of the discharge.

Reply: 2-3 TBq year-1 or 5–8 GBq day-1 from both FNPP1 and FNPP2 are reported value by TEPCO to the government in annual reports to the commission overseeing the operational status and environmental impact of nuclear power plants written in Japanese.

L313-316 "After the FNPP1 accident,⋯ no difference was observed before the accident among the monitoring stations."

After the FNPP1 accident, relatively lower tritium values suddenly appeared on Fig. 8. I guess that analytical procedure during the year periods of 2011-2019 seemed to be different from before 2010. The authors should explain that. I could not find out where those values were observed. The authors should indicate sampling points.

Reply: The data in figure 8 represent a large set of data but cannot be adequately processed to meet an international journal standard due to information gaps. Therefore Fig8 and associated sentences in the main text lines 307-317 will be deleted.

L323 "0.05 m s$^{-1}$"

How did the authors obtain this rate?

Reply: One key issue around 56N of FNPP1 is there is strong coastal flow as indicated in Fig.2 in Tsumune et al., 2013 Biogeoscience. The net speed of V-component is -0.048 m s-1 as an average in 2014 so we took into account the value of 0.05 m s-1.for flux calculation in this study. This speed corresponds to 4 km day-1 movement to the south. Therefore, to maintain an order of magnitude higher 3H activity compared to surrounding waters at 5-6N of FNPP1, it was necessary to set up an order of magnitude higher flux from FNPP1.

L323-325 "freshwater flux from the rivers … along the Fukushima coast."

References concerning river discharge are necessary. Sakuma et al. (2019) in Table A2? If so, please indicate this on the manuscript. Please explain how to calculate the fluxes. From parameters on Table A2, it is hard to understand. Also, I could not find out how the authors obtained those parameters.

Reply: The river flow was calculated according to the area of the watershed (km2) x the amount of monthly precipitation (mm month-1) x the part of the flow to the ocean (assumed to be 0.6) as shown in the Supplementary Table 2. The watershed area (km2) of each river was quoted from Sakuma et al., 2019 as you mentioned. My estimate is a simple estimate compared to a reservoir model estimate by Sakuma et al., 2019, but the freshwater flow estimated by both methods showed good agreement.

L327-329 "We obtained the estimates for June 2014 and October 2014, and also with very heavy rains encountered in October…. are listed in Table 3"

Please describe more detailed explanation for the estimation of flux.

Reply: OK, we will add more details for the estimation of flux.

L332 "(Kanda 2012)"
This is not listed in references.
Reply: Ok, we will add this in the reference list.

L333 "Monoageba"
?
Reply: This is the name of the point in the FNPP1 port

L336-339 "In contrast, considering …. $^{3}$H flux from the FNPP1 site was found to be 28 to 86 GBq day$^{-1}$ as shown in Table 3,"

Please describe more detailed explanation for the estimation of flux for readers.

Reply: In Tsumune et al., 2020, there is an equation to estimate flux from the FNPP1 site based on the activity concentration at **56 discharge outlet and south discharge outlet** as F(t) = C(t) /5E-8. So, we can estimate the flux from the FNPP1 site based on the observed 3H activity concentration at **56 discharge outlet and south discharge outlet**.

L340-341 "One of the reasons … and the port of FNPP1."

I could not understand what the authors explain. Please explain more detail. I think this topic is the most important in this study.

Reply: Kanda's method seeks flux into the port. Cs-137 concentrations at 56 discharge outlet and south discharge outlet are always lower than the ones in the port. The dilution rates inside and outside the port are different. Therefore, the relationship between flux and in-port concentration is different from the relationship between flux and out-of-port concentration

at 56 discharge outlet and south discharge outlet.

In April/May 2011, however, both flux estimations from the activity of inside the port and from the activity of 56N showed excellent agreement. This point is also an important issue. But we don't know why flux to outside of port is one order of magnitude larger than the one into the port as shown in this study period as in 2014 and 2019.

There is no question that the two methods are different, but when they differ by an order of magnitude, further study is needed.

L342 "Conclusion"

I do not think this section is conclusion.

Reply: Ok, we will revise the conclusion as below.

From our dataset, it appears clearly that open-water levels corresponding to background levels were encountered at a distance of around 15km from the coast. Within this area, the impact of accidental releases does not appear clearly and the 3H levels seem to be linked to the dilution of river inputs since the levels in freshwater can be up to an order of magnitude higher than in open marine waters. However, some values in rivers could suggest a contribution from the FNPP1 accident may be via groundwater inputs but this possible contribution seems to be quite low at least within our dataset. However when taking into account values obtained in the discharge canal from reactors 5 and 6 North outlet canal of the FNPP1 a clear contribution from the accident can be seen with fluxes on an order of magnitude higher compared to fluxes arising from 9 rivers located just north (7 rivers) and south (2 rivers) of the FNPP1 site.

Due to the variability of 3H background levels in rivers due to various factors such as the residence time of water masses at the scale of the watershed, size of watershed, season, latitude, etc.. further comprehensive study is required to reflect this variability in the near coastal area close to the damaged FNPP1.

L416 "References"

Please check the reference style. The number of 137Cs or 134Cs should be superscript. Also please check references whether these are cited in the main body. There are papers written by another language.

Reply: Ok, we will do it. As you pointed out, Tagomori et al., 2015 is written in Japanese. But this article is important for the discussion at section 4.2, therefore the authors asked Mr. Tagomori to publish the data in the document as a dataset with EN abstract with DOI.

L518-520
?
Reply: We delete this reference because we delete the sentences where it was cited.

End of reply to RC1.

---

## Author Comment (AC2)

**Replies to Comment on bg-2021-10 by anonymous referee #2**

**Dear anonymous referee #2,**

**We submit our replies to your comments as below.**
**Thank you very much for your comments which greatly contribute to the improvement of our manuscript.**

**Best regards,**

**Michio**
* * *
Reply to Referee#2:

This was a difficult paper to read/review due to the lack of flow and logical sequencing in the paper. Moreover, the grammar and writing style made it difficult to interpret the precise intent of the authors. The paper does not appear to have been proof-read carefully – there are duplicate redundant sentences (e.g., lines 130-132, 138-139) and figure labels do not always match what is in the figure caption (e.g., Figure 4).

Reply: In some parts of the main text, there are obvious redundant sentences as RC2 pointed out. The Y-axis label in figure 4 should be 3H, not 137Cs. We will recheck all the text, tables and figures again before submitting a revised version.

The data, 3H and 137Cs data collected over 2014-2018 from the coastal waters of eastern Japan near the Fukushima Nuclear Power Plant and two sites further south, seem to be looking for a home. Although Biogeosciences hosted a special issue dedicated to the Fukushima event, it is unclear to me if Biogeosciences is the 'best' journal for these newer monitoring style data compared to a journal more specific to radionuclides and radiochemistry (e.g., Journal of Environmental Radioactivity, Journal of Radioanalytical and Nuclear Chemistry).
Reply: Since this article is not only dedicated to radionuclide activity, but also to their dynamics including the issues of freshwater fluxes, Biogeosciences appears very appropriate.

The authors expend a good deal of writing for background on atmospheric weapons testing derived tritium ($^3$H). I do not believe this was a good use of space in the manuscript. Of the $^3$H produced by weapons testing, and if we use 1963 as the initial time zero, less than ~6% of weapons testing $^3$H is still in the environment (atmosphere, terrestrial, ocean reservoirs). The authors could significantly shorten and tighten the introduction to simply state the background $^3$H (and $^{137}$Cs) in the western subtropical Pacific, that controlled releases from FNPP elevated coastal water $^3$H prior to the earthquake/tsunami induced cataclysm, and then go straight into line 120: "In this paper, we present…"
Reply: As you pointed out as well as referee#1, we will shorten the introduction.

The authors could also state the purpose or what they were looking to explore/understand. Were they looking to better understand/constrain the relative influences of FNPP impacted

submarine groundwater discharge versus surface (river) input of $^3$H and $^{137}$Cs on coastal water concentrations?

Reply: We revised the introduction and stated the objectives of this article more clearly at the end of the introduction as below.

In this paper, we present results of the $^3$H activity concentration observed during the SoSo 5 rivers cruise and at the Tomioka port and Hasaki, a pier of the Hasaki Oceanographical Research Station of the Port and Airport Research Institute, and discuss the behaviour of $^3$H in the coastal region of Fukushima. An assessment of various contributions to both $^3$H and 137Cs fluxes into the coastal area close to the FNPP1 site was carried out taking into account i) riverine flux of these radionuclides based on precipitation amount on the catchments of several rivers, ii) fluxes estimation using from the FNPP1 site by previously reported method (Kanda 2012, Tsumune et al., 2012, 2013, 2020) and iii) fluxes from open water towards based on the speed of the coastal current and the activity concentrations characterizing these open water. All of these results are discussed together with 3H activity concentrations in river and open sea waters already published.

Does the different physical chemistry of cesium and tritium lead to different input functions in the coastal waters (eg., cesium will desorb off particles when it gets to higher salinity)? This is particularly relevant with regards to submarine groundwater discharge which is a significant source of 137Cs (e.g., Sanial et al., 2017 www.pnas.org/cgi/doi/10.1073/pnas.1708659114) post direct discharge (eg., Buesseler et al, 2012).

Reply: RC2's comment that "submarine groundwater discharge which is a significant source of 137Cs (e.g., Sanial et al., 2017 www.pnas.org/cgi/doi/10.1073/pnas.1708659114) post direct discharge (eg., Buesseler et al, 2012)". We do not find such impact when we analyzed the data around the coastal region of Fukushima. Although Sanial et al., 2017 presented a very interesting phenomenon, it was put into evidence at a specific area south of FNPP1 site where waters were highly contaminated by initial direct release. Their estimates might be overestimated because most of the area is rocky. We observed that 137Cs and 3H activity concentration along the Fukushima coast showed a maximum at FNPP1 site with decreasing activities both north and south of FNPP1 site. If the flux from the contaminated beach was significant at the time of our sampling, the distribution of 137Cs activity concentration along the coast should show a different shape.

Key takeaways:

From the TEPCO 56N canal data, it is pretty clear that FNPP is (still) a source of 3H, regardless of the sensitivity of their methods being limited to > 1650Bq-m-3.
Reply: Yes, it is.

The Aoyoma et al., additional data capture the input of 3H and 137Cs into coastal waters.
Reply: Yes, this point is one of key issues of this article. Thanks,.

One of the most intriguing aspects of the data is the 3H/137Cs ratio that has varied post direct discharge in 2011 to the newer data. The authors do not provide a credible discussion/interpretation of this observation.
Reply: Ok, both 3H/137Cs activity ratio and their respective activities during the period 2012 -2016 are showed in figures below. 3H/137Cs activity ratio increase over time as mentioned in our text and not decline, while activity concentrations of 3H and 137Cs at 56N, shows a different decreasing trend.

[Figure]

In the current text, we stated "During the period from 2013 to 2016 at 56N of FNPP1, the 137Cs activity concentration decreased around two orders of magnitude lower due to decontamination effort of TEPCO while the 3H activity concentration decreased gradually, then the 3H/137Cs activity ratio tended to increase from 1 to 10 during the period from 2013 to 2016 (Figure are not shown). " Only we need to do here is to revise from "two orders" to "one order". We keep the other part of the sentence because it is the correct description of the trend of the two radionuclides. Since the observation at 56N was carried out very close to the source, we can assume that the observed trend was little affected by the water movement but mostly represents the change in the characteristics of the source. If the observation was done after some movement, salinity may affect the ratio as RC1 pointed, but this is not the case here.

So, in the revised main txet, this part will be as below;
Between 2013 and 2016 at 56N of FNPP1, the 137Cs activity concentration decreased by about one order of magnitude due to decontamination effort of TEPCO while the 3H activity concentration decreased only gradually, leading to an increase of the 3H/137Cs activity ratio from 1 to 10 (Figure are not shown).

What are the uncertainties on the flux (input) estimates? Are there any 'real' differences in the estimates provided in eg., table 3?
Reply: The uncertainties on the flux budget depend on the uncertainties of each parameter entering into r the calculation.
For the activity of 3H and 137Cs in open water, uncertainties are around 10-20%. For the river waters at Ukedo and Tomioka, uncertainties are around 10 %. For the activity of 3H and 137Cs at 56N of FNPP1 and in the port of FNPP1, variability is large and uncertainties are around 50 -100 %. Therefore, the order/rank of estimated values is real. Therefore, in the main

text, we thought it safe to say that "Regarding the 3H fluxes, the largest source comes from the open-water inflow from the north of FNPP1, with 52 GBq day-1 while the rivers north of FNPP1 contribute 3–6 GBq day-1. From the port of FNPP1, we used Kanda's method (Kanda 2012), ,,,, that led to 3H fluxes in the range of 1.9–4.5 GBq day-1 in three cases in 2014 and 2019, which is comparable with the 3H fluxes from the rivers located north of FNPP1." In other words, our estimate is at least correct as an estimate of the relative importance of the different contributions.

End of reply to RC2.

---

## Author Comment (AC3)

**Additional reply to Comment on bg-2021-10 by anonymous referee #1**

**Dear anonymous referee #1,**

**We already submitted our replies to your comments on 27 August 2021. Today, we add a few regarding the tritium activity concentration in monthly precipitation at Tamura, Fukushima in 201-2014 as blow.**

**Best regards,**

**Michio**
**\*\*\*\*\*\*\*\*\*\*\*\*\*\*\*\*\*\*\*\*\*\*\*\*\*\*\*\*\*\*\*\*\*\*\*\*\*\*\*\*\*\*\*\***

L266 "Tamura-shi"

Same as above comment
Reply: We put a mark on Tamura-shi in the revised figure 1. And the exact location of the sampling site is shown in "Tritium concentration in precipitation at Tamura City, Fukushima in 2013-2014, Tagomori et al., doi: 10.34355/KEEA.00086." which is published on 3 September 2021.

L416 "References"

Reply: As you pointed out, Tagomori et al., 2015 is written in Japanese. But this article is important for the discussion in section 4.2, therefore the authors asked Mr. Tagomori to publish the data in the document as a dataset with EN abstract with DOI. And the 3H activity concentration dataset at Tamura, Fukushima in 2013-2014 was published as "Tritium concentration in precipitation at Tamura City, Fukushima in 2013-2014, Tagomori et al., doi: 10.34355/KEEA.00086." So everybody can cite this.

End of additional reply to RC1.